# Elo Uncovered: Robustness and Best Practices in Language Model Evaluation

**Meriem Boubdir**
Cohere For AI
meri.boubdir@gmail.com

**Edward Kim**
Cohere
edward@cohere.com

**Beyza Ermis**
Cohere For AI
beyza@cohere.com

**Sara Hooker**
Cohere For AI
sarahooker@cohere.com

**Marzieh Fadaee**
Cohere For AI
marzieh@cohere.com

## Abstract

In Natural Language Processing (NLP), the Elo rating system, originally designed for ranking players in dynamic games such as chess, is increasingly being used to evaluate Large Language Models (LLMs) through "A vs B" paired comparisons. However, while popular, the system's suitability for assessing entities with constant skill levels, such as LLMs, remains relatively unexplored. We study two fundamental axioms that evaluation methods should adhere to: **reliability** and **transitivity**. We conduct an extensive evaluation of Elo behavior across simulated and real-world scenarios, demonstrating that individual Elo computations can exhibit significant volatility. We show that both axioms are not always satisfied, raising questions about the reliability of current comparative evaluations of LLMs. If the current use of Elo scores is intended to substitute the costly head-to-head comparison of LLMs, it is crucial to ensure the ranking is as robust as possible. Guided by the axioms, our findings offer concrete guidelines for enhancing the reliability of LLM evaluation methods, suggesting a need for reassessment of existing comparative approaches.

## 1 Introduction

In the rapidly evolving field of Natural Language Processing (NLP), the task of accurately and reliably evaluating LLMs has become increasingly challenging [32, 10, 50, 26, 43]. Human feedback has emerged as an indispensable tool in this performance assessment process, serving as a qualitative metric that captures nuances that automated scoring mechanisms often fail to address [2, 3, 4, 50, 13, 12]. These human-centered evaluations, highly valuable to the overall progress of the NLP field, typically adopt an *"A vs B"* comparative setup, turning evaluations into a zero-sum game between language models. Pairwise comparisons, however, are fundamentally difficult to scale for large pools of models, due to the quadratic growth of comparisons required. Fortunately, this paired feedback structure [56] naturally lends itself to the Elo rating system, originally designed for ranking chess players (including those who have never before played each other) for better matchmaking [16].

Under the Elo rating system, players' skills are indicated by an *Elo rating*, where higher ratings indicate higher skill, and all players can be ranked best to worst using this scalar Elo rating. In the standard formulation (see Section 2), a player rated at 1800 has $10 : 1$ odds of winning against a player rated at 1400. After a match, the winner takes rating points from the loser in a zero-sum fashion [16]. Thus, with the Elo rating system, we can efficiently integrate subjective human feedback on paired *"A vs B"* language model completions into a structured and unified rating system to assess the performance of language models.

38th Conference on Neural Information Processing Systems (NeurIPS 2024).

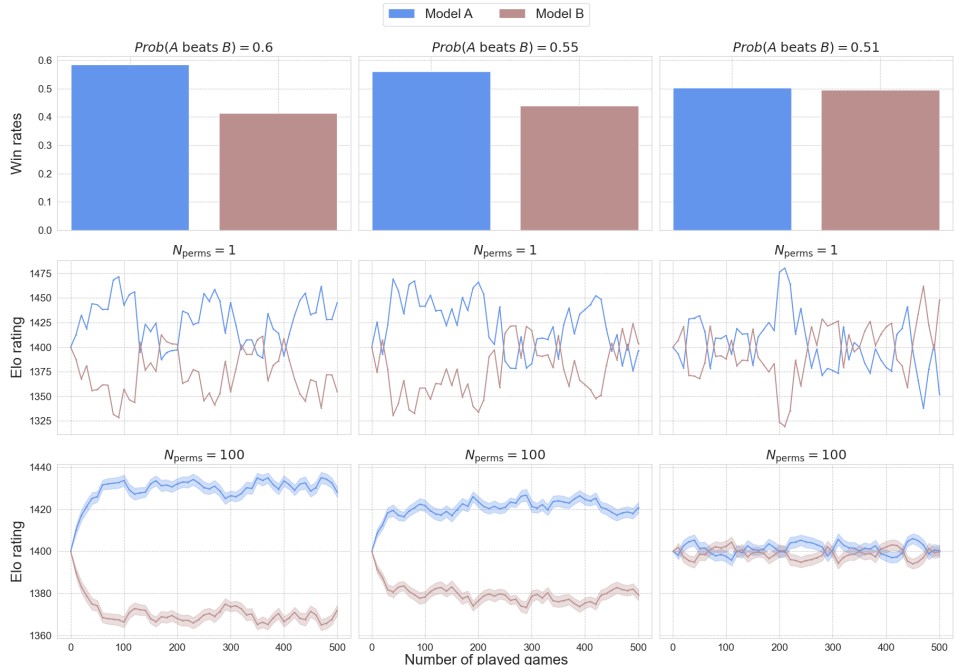

Figure 1: **Impact of win probabilities and permutation sampling on Elo ratings**: Comparing Model A and Model B across three different win probabilities ($Prob(A$ beats $B) = \{0.6, 0.55, 0.51\}$) with two levels of permutation sampling ($N_{\text{perms}} = 1$ and $N_{\text{perms}} = 100$). The top row displays the observed win rates, the middle one the Elo ratings with a single permutation, and the bottom one the mean and standard error of the mean (SEM) of Elo ratings across 100 permutations.

The core principles of Elo rating have proven to be resilient and adaptable due to its dynamic adjustments, relative rating focus, consistency across skill levels, and simplicity and transparency. As a result, the Elo rating system has found diverse applications, from predicting sports events outcomes [8, 25, 31, 53], and facilitating matchmaking in massively multiplayer online games like StarCraft II and Dota [15, 44, 35, 17], to its recent use in the evaluation of LLMs [2, 3, 4, 50, 13, 12, 54, 33]. However, to-date there has not been a comprehensive examination of the compatibility of Elo scores and LLMs evaluation.

Unlike dynamic competitors that evolve over time, LLMs have static capabilities and operate in a time-agnostic context. In this setting, evaluations of LLMs are not constrained by a preset number of turns, as is the case with tournament timelines or predefined match sequences. Moreover, the ordering of matches can significantly influence the final Elo scores and, consequently, model rankings. This oversight is particularly concerning, given the direct impact of Elo system rankings on both research directions and real-world applications in NLP as well as its widespread adoption [2, 3, 55, 57, 29, 4, 50, 13, 12, 54, 33].

This study aims to close this research gap by adopting an axiomatic approach and scrutinizing both the reliability and limitations of the Elo rating system when applied to LLMs. We study two fundamental axioms that evaluation methods should adhere to: **reliability** and **transitivity**. Through theoretical and empirical analyses grounded in collected human feedback data, our contributions provide a comprehensive understanding of when and how to reliably employ the Elo system for LLM evaluation, thus offering valuable guidelines for researchers and practitioners in the NLP field.

We find that Elo ratings for LLMs are highly sensitive to the *order of comparisons* and the choice of hyperparameters. Moreover, desirable properties such as transitivity are not always guaranteed and can be unreliable unless there is comprehensive human feedback data for all *unique pairwise comparisons* among models in the feedback pool. The sensitivity of Elo ratings becomes more pronounced when dealing with models that exhibit *similar* performance levels. We illustrate the best practices for addressing Elo rating sensitivities by offering guidelines for hyperparameter selection and matchmaking scenarios.

**Implications of our work** As LLMs rapidly advance, evaluation leaderboards are gaining popularity to assess the performance of newly introduced models using Elo scores. Elo can also be used in the learning framework of LLMs to produce a ranking of models and their outputs for preference training. No research has explored the nuances of using Elo scores to compare LLMs, which, unlike chess, exhibit static capabilities and operate in a time-agnostic manner. We show that Elo rating does not always satisfy two critical axioms—reliability and transitivity—leading to rankings of models that are not accurate. Our research offers guidelines for reliable and robust implementation of Elo scores when comparing LLMs. Deviation from our recommendations could result in inaccuracies when ranking LLMs, particularly in situations where model performances are closely matched, and Elo score differences are minimal (a common occurrence in many real-world scenarios).

## 2   Elo Algorithm Explained

We provide the standard mathematical formulation of the Elo algorithm [16], contextualized to the setting of LLM evaluation. In this formulation, let $\mathcal{M}$ be a set of models, and each model $i \in \mathcal{M}$ is assigned an initial numerical Elo rating $R_i$. For each match between two models, we calculate the *expected score*, then update the *ratings* of both models as follows:

### 2.1   Expected Score Computation

For a given paired zero-sum match-up between two models $A$ and $B$ ($A, B \in \mathcal{M}$), each with respective pre-match ratings $R_A$ and $R_B$, the expected scores $E_A$ and $E_B$ (i.e., match outcomes) are computed as:

$$E_A = \frac{1}{1 + 10^{(R_B - R_A)/400}} \quad \text{and} \quad E_B = \frac{1}{1 + 10^{(R_A - R_B)/400}} \tag{1}$$

In this context, the factor of $400$ [16] precisely adjusts the sensitivity of the expected score to differences in ratings. A $400$-point advantage in ratings translates to a $10 : 1$ odds in favor of the higher-rated model, providing an interpretable metric for performance comparison. For evenly matched models ($R_A = R_B$), both $E_A$ and $E_B$ equate to $0.5$, reflecting a $50 : 50$ win probability for both models.

### 2.2   Rating Update Mechanism

Following each match, the Elo ratings are updated based on the observed win-loss outcome. The rating adjustment for each model is dictated by the equation:

$$R'_A = R_A + K(S_A - E_A) \tag{2}$$

Here, $S_A$ represents the actual score achieved by model $A$, which can take on either the value 0 for a loss or 1 for a win. Model B's Elo rating is updated via the same method. The $K$-factor serves as a variable hyperparameter to adapt the rate of change in rating to different scenarios. A higher $K$-factor results in larger changes in the Elo score after each match-up, making the scoring more sensitive to individual results. A lower $K$-factor, in contrast, makes the Elo ratings more stable, with smaller changes after each match. In chess, the $K$-factor is usually set to 16 for masters and to 32 for novice players.

## 3   Desirable Properties of Elo

The objective of using Elo scores to rank models is to establish a comparative understanding of the performance hierarchy among them. When incorporating a new model into an already ranked list, only a limited number of pairwise annotations are required to determine its position in the ranking. The ability to infer a model's relative performance compared to all previous models in the list relies on the robustness of the scoring method and the transitive property of the ranking system. We describe these desirable properties through two axioms: *transitivity* and *reliability*.

### 3.1 Axiom 1: Transitivity

A desirable property of any rating system is transitivity because it ensures consistency and logical coherence in how entities are ranked or rated. Transitivity in this context means that if player $A$ beats player $B$, and player $B$ beats player $C$, then player $A$ is expected to beat player $C$. If the ranking of large language models exhibits transitivity, we can deduce their comparative performance without the need for direct head-to-head evaluations between every pair of models. The central assumption in developing various leaderboards for comparing language models is that the rankings adhere to the principle of transitivity [57].

While Elo's design inherently assumes transitivity, our synthetic data which are derived from realistic scenarios, uncovers certain circumstances that violate this assumption. Such anomalies can affect the final ranking of language models and their relative performance assessments.

### 3.2 Axiom 2: Reliability

We consider two aspects of reliability:

**Sensitivity to ordering:** Unlike chess or time-bound sports where match sequences are structured, in LLM evaluations all matches can occur independently and in parallel, amplifying the sequence's influence on final model ranking. In this context, each match represents the performance comparison between two models on a specific prompt. If the prompts are presented in a specific order, and one model happens to perform better on the initial set of prompts, it may gain an advantage in subsequent comparisons due to the cumulative effect of its early success. This inherent variability prompts us to investigate the extent to which match-up ordering affects the robustness of Elo ratings.

**Sensitivity to hyperparameters:** The sensitivity of hyperparameters can compromise the robustness of Elo scores leading to inconsistent rankings. Evaluating and understanding this sensitivity is crucial for building evaluation frameworks that maintain consistency across diverse models. In this work, we evaluate the sensitivity of Elo performance to one key hyperparameter, the $K$-factor. This factor acts as a scaling constant in the Elo rating system, pivotal for updating ratings after each matching. It essentially determines how quickly a model's rating converges to what can be considered its "true" skill. While conventional applications like chess use standard $K$-factor values, these may not be directly applicable in the context of evaluating LLMs due to the unique characteristics and requirements of this domain.

## 4 Synthetic Human Feedback

Given the costly and time-consuming nature of human evaluations, studying the Elo system's behavior under various scenarios becomes challenging. To circumvent these limitations, we first validate the properties of Elo using synthetic data generation via Bernoulli processes to simulate various human feedback scenarios. In Section 6 we extend these evaluations to include real-world human feedback. This time-agnostic and independent setup of LLM evaluations resembles a Bernoulli process [6], a sequence of independent experiments, each yielding a simple "win" or "loss" outcome, representing one model outperforming another. We use this setting to control the characteristics of the distribution and evaluate the different desirable properties of a rating system.

In this controlled setting, our primary objectives include testing the **transitivity** axiom—whether a consistently higher-rated model outperforms those with lower ratings in all scenarios. Additionally, in studying the **reliability** axiom, we explore how the Elo scores are affected by the *order in which models are compared* and the sensitivity to *hyperparameter adjustments*, particularly the $K$-factor. This synthetic setup offers a robust platform to dissect and understand the dynamics of the Elo rating system in the context of LLM evaluations, without the constraints and limitations of relying solely on real-world human feedback.

### 4.1 The Bernoulli Analogy

Pairwise comparisons in LLM evaluation draw parallels with the foundational principles of the Bernoulli experiment in probability theory. This section studies the similarity between human feedback-based evaluations and the Bernoulli experiment's principles.

**Preliminaries**    A Bernoulli trial is a random experiment with exactly two possible outcomes, "success" or "failure". The outcomes adhere to the probability condition:

$$P(\text{success}) + P(\text{failure}) = 1 \tag{3}$$

Here, the random variable $\mathcal{X}$ denotes the outcome, where $\mathcal{X} = 1$ implies success, and $\mathcal{X} = 0$ signifies failure. The probabilities associated with these outcomes are given by:

$$P(\mathcal{X} = 1) \quad = p, \quad P(\mathcal{X} = 0) = 1 - p \tag{4}$$

with $0 \leq p \leq 1$, the "success" probability.

**Mapping to Human Feedback**    When comparing two models, $A$ and $B$, across $N$ pairwise evaluations, the setup aligns with a Bernoulli process. This process comprises a sequence of independent and identically distributed (*i.i.d*) Bernoulli trials. To frame this analogy, we designate a win probability, $P(A_{\text{win}})$, to model $A$. Leveraging a Bernoulli random variable, $\mathcal{X}$, as a means to simulate synthetic human feedback, we proceed as follows:

1. A sample is drawn from $\mathcal{X}$ using $P(A_{\text{win}})$.
2. If $\mathcal{X} = 1$, feedback suggests a preference for model $A$.
3. Otherwise, model $B$ is favored.

**Extending to Multiple Players**    Given a finite set of $n$ distinct models $\mathcal{M}$, their pairwise comparisons can be formulated as:

$$\binom{n}{2} = \frac{n!}{2!(n-2)!} = \frac{n(n-1)}{2} \tag{5}$$

This formula yields $\binom{n}{2}$ unique pairs $(A, B)$ where $A, B \in \mathcal{M}$ and $A \neq B$. For each pair, a Bernoulli process comprising multiple Bernoulli experiments is conducted to discern which model performs better over a sequence of trials.

## 4.2    Synthetic Data Generation

Building upon the Bernoulli process analogy, when conducting multiple independent evaluations between two models, the distribution of the number of times one model is preferred over the other naturally follows a binomial distribution. For $N$ pairwise comparisons, the relation is:

$$P(k; N, p) = \binom{N}{k} p^k (1-p)^{N-k} \tag{6}$$

where $P(k; N, p)$ is the probability of one model being preferred $k$ times out of $N$ evaluations. $p$ is the success probability and $\binom{N}{k}$ is the binomial coefficient, representing the number of ways to choose $k$ successes from $N$ trials.

## 5    How Robust Are Elo Scores?

This section describes rigorous stress tests designed to investigate whether the two axioms, presented in Section 3, are satisfied in this evaluation framework. We focus on critical desirable properties of a ranking mechanism – that it should (1) be insensitive to match-up ordering, (2) not be overly sensitive to hyperparameters like the $K$-factor, and (3) preserve properties of transitivity. Subsequently, we provide empirically grounded guidelines for a safe and interpretable application of Elo ratings.

### 5.1    Impact of Ordering on Elo Ratings

**Experimental Setup**    To quantify the effect of match-up ordering, we generate a baseline sequence of $N_{\text{games}} = 1000$ match outcomes between models $A$ and $B$ (see Equation 6), reflecting the scale typical of LLM evaluations via human feedback. We hold $N_{\text{games}}$ constant for the entirety of our study to maintain consistency. From this baseline, we derive $N_{\text{perms}}$ distinct permutations, each involving a complete reshuffling of the initial sequence to simulate various chronological orders in which the games might unfold. It is important to note that we are not generating new match outcomes for each permutation; instead, we simply reorder the existing data to explore the potential impact of different

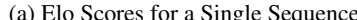

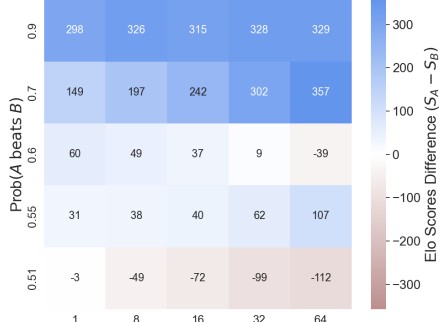

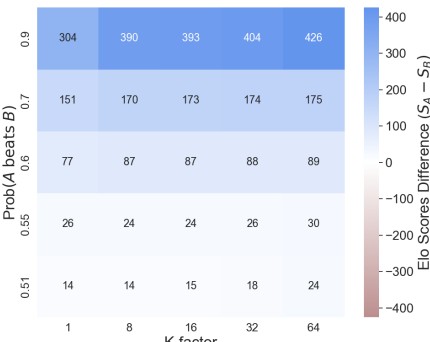

(a) Elo Scores for a Single Sequence

(b) Elo Scores Averaged Over 100 Permutations

Figure 2: Final Elo scores difference ($S_A - S_B$) as a function of $K$-factor and $N_{\text{perms}}$. Positive values reflect the expected ranking where Model $A$ is superior to Model $B$, while negative values indicate a discrepancy, falsely suggesting that Model $B$ has a higher Elo score than Model $A$. We compare between a single sequence of outcomes and averages over $N_{\text{perms}} = 100$ unique permutations.

match-up sequences. For each reordered sequence, we update the Elo ratings $R_A$ and $R_B$ according to equation 2, resetting both ratings to an initial value of 1400 at the start of each permutation. Finally, we compute average Elo ratings per match across all $N_{\text{perms}}$ permutations, ensuring a robust analysis that takes into account the full range of possible match-up orders.

We repeat this process to generate baseline sequences and their respective reorderings for a set of selected winning probabilities enabling us to inspect ratings' behavior under various real-world scenarios. $N_{\text{perms}}$ is varied from a minimum of 1 to a maximum of 10k, providing a robust sample size for statistical analysis (see Figure 3). Subsequently, we compute the average Elo ratings per match across all permutations. These averages, $\bar{R}_A$ and $\bar{R}_B$. particularly for $N_{\text{perms}} = 1$ and $N_{\text{perms}} = 100$, are visualized to offer insights into the stability of the ratings, as shown in Figure 1.

**Key Findings** Our analysis underscores the interplay between winning probability $P(A_{\text{win}})$ and the number of different orderings $N_{\text{perms}}$ on the stability of Elo ratings after each update. For $P(A_{\text{win}}) \geq 0.6$, Elo ratings demonstrate high stability; additional results for $P(A_{\text{win}}) = 0.65$ and beyond are available in Appendix B. On the other hand, for $P(A_{\text{win}}) \approx 0.5$, ratings exhibit significant instability for a single sequence. As depicted in Figure 1, when both models have win probabilities around $0.5$, Elo ratings frequently intertwine, making it challenging to discern a clear performance difference between the two. The instability plateaus as $N_{\text{perms}}$ exceeds 100, resulting in stabilized Elo ratings that align closely with the preset winning probabilities. For instance, at $P(A_{\text{win}}) = 0.55$, the average Elo rating for Model $A$, $\bar{R}_A$, consistently exceeds that for Model $B$, $\bar{R}_B$,

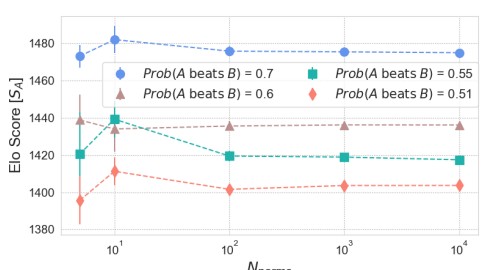

Figure 3: Variation of Model A's average Elo score with increasing number of permutations $N_{\text{perms}}$ for different probabilities of Model A winning ($P(A \text{ beats } B)$). Error bars indicate standard errors of the mean.

when averaged across multiple permutations, reflecting an accurate performance-based ranking of these models. These observations validate our concerns highlighted earlier, emphasizing the critical role of $N_{\text{perms}}$ for a reliable interpretation of Elo ratings in LLM evaluations. In Elo-based evaluations, the sequence of model comparisons can significantly influence the final Elo scores, particularly in scenarios with models of similar quality, where this effect is magnified.

## 5.2 Sensitivity to Hyperparameters

**Experimental Setup** We extend our previous approach by conducting tests across a range of winning probabilities and multiple $K$-factor values $(1, 8, 16, 32, 64)$. We compute and compare the

Table 1: Investigation of Elo score reliability in capturing true model hierarchies across varying configurations. Scenarios explore the transitive relationship $A > B$ and $B > C \implies A > C$. The star (*) indicates cases where the Elo score fails to accurately reflect the expected hierarchy of models. $\approx$ represents models with similar performance; $\gg$ indicates that a model significantly outperforms the other one.

| Scenario | Model | Elo-based Models Ranking per Configuration | | | |
|---|---|---|---|---|---|
| | | $N = 1, K = 1$ | $N = 100, K = 1$ | $N = 1, K = 16$ | $N = 100, K = 16$ |
| ♛ | $A$ | 1539.43 | 1528.50 ± 0.35 | 1650.93 | 1584.78 ± 3.09 |
| $A \gg B$ | $B$ | 1390.47 | 1410.33 ± 0.54 | 1381.17 | 1406.48 ± 3.23 |
| $B \gg C$ | $C$ | 1270.10 | 1261.17 ± 0.33 | 1167.90 | 1208.74 ± 2.71 |
| ♜ | $A$ | 1502.09 | 1495.92 ± 0.36 | 1509.08 | 1526.04 ± 3.03 |
| $A \gg B$ | $B$ | 1337.48 | **1342.70\*** ± 0.53 | 1379.00 | 1340.83 ± 2.83 |
| $B \approx C$ | $C$ | 1360.42 | **1361.38\*** ± 0.38 | 1311.92 | 1333.13 ± 2.68 |
| ♝ | $A$ | 1437.97 | **1433.84\*** ± 0.41 | 1440.31 | 1460.22 ± 2.90 |
| $A \approx B$ | $B$ | 1455.10 | **1453.84\*** ± 0.61 | 1481.04 | 1452.87 ± 3.25 |
| $B \gg C$ | $C$ | 1306.93 | 1312.32 ± 0.34 | 1278.65 | 1286.91 ± 2.72 |
| ♞ | $A$ | 1426.33 | 1419.73 ± 0.36 | 1407.44 | 1432.26 ± 2.93 |
| $A \approx B$ | $B$ | 1390.47 | 1393.29 ± 0.59 | 1386.17 | 1392.75 ± 3.04 |
| $B \approx C$ | $C$ | 1383.20 | 1386.99 ± 0.41 | 1406.39 | 1374.99 ± 3.12 |

average Elo scores $\bar{S}_A$ and $\bar{S}_B$ for $N_{\text{games}} = 1000$ and $N_{\text{perms}} = \{1, 100\}$. The differences between these final averages for Model $A$ and Model $B$ are summarized in Figure 2 to assess the stability and expected ranking between the two models.

**Key Findings** As shown in Figure 2, notable instability is observed in model rankings based on the final Elo scores when we consider a single sequence of paired comparisons (i.e., $N_{\text{perms}} = 1$), especially for winning probabilities nearing 0.5. This instability is markedly exacerbated at higher $K$-factors. In contrast, the picture changes when coupling higher $K$-factors with raising the number of permutations to at least 100. Higher $K$-factors, in this multi-permutation scenario, speed up the differentiation between models' Elo scores, enabling faster convergence to their true skill levels. This yields much more stable and reliable model rankings. It is noteworthy that this faster convergence is observed to be more reliable for higher winning probabilities, which corresponds to skewed win rates in a real-world scenario.

### 5.3 Transitive Properties of Elo Scores

**Experimental Setup** The transitivity property of the Elo scores is defined as:

$$A > B \quad \text{and} \quad B > C \implies A > C \tag{7}$$

To test the transitivity property, we design four distinct scenarios that model real-world conditions:

- ♛ Model $A$ beats model $B$ and model $B$ beats model $C$ both with high win probabilities ($P_{\text{win}} = 0.75$).
- ♜ Model $A$ beats model $B$ with a high win probability ($P_{\text{win}} = 0.75$), model $B$ beats model $C$ with a win probability close to 0.5 ($P_{\text{win}} = 0.51$).
- ♝ Model $A$ beats model $B$ with a win probability close to 0.5 ($P_{\text{win}} = 0.51$), model $B$ beats model$C$ with a high win probability ($P_{\text{win}} = 0.75$).
- ♞ Model $A$ beats model $B$ with a win probability of 0.54, model $B$ beats model $C$ with a win probability of 0.51.

In each of these scenarios, we simulate matches for paired comparisons "$A$ vs. $B$" and "$B$ vs. $C$" and then rearrange these matches in an arbitrary order to form our baseline sequence. This approach mimics how Elo ratings are computed for online leaderboards in the evaluation of large language models [54, 33]. We then analyze whether Elo scores maintain the expected model hierarchies.

**Key Findings** The outcomes from all four scenarios, detailed in Table 1, demonstrate the performance of Elo-based rankings across various configurations. In scenarios where there is a clear disparity between models (e.g.,♛), Elo ratings accurately reflect the expected hierarchy. However, in

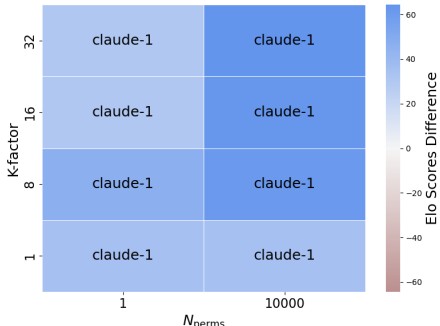 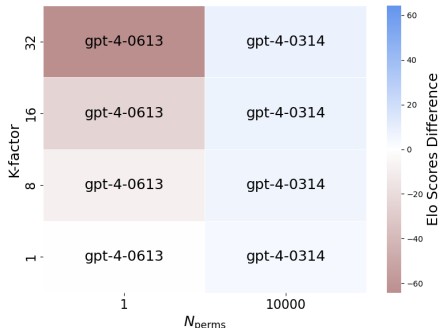

(a) Experiment: Claude-1 vs. Claude-2.1
**Recorded Win rates**: $\boxed{0.59 \text{ vs } 0.41}$

(b) Experiment: GPT-4-0314 vs. GPT-4-0613
**Recorded Win rates**: $\boxed{0.51 \text{ vs } 0.49}$

Figure 4: Elo score differences ($S_A - S_B$) across varying K-factors and $N_{\text{perms}}$. Positive values in the heatmap indicate that the expected ranking is maintained (Model A outperforming Model B), while negative values suggest a ranking inversion, where Model B appears to outperform Model A, contrary to the actual win rates. Each cell's label indicates the model with the higher Elo score.

more complex cases such as ♖ and ♗, where one model significantly outperforms a second, which in turn is closely matched with a third, the rankings become less stable, challenging the assumption of transitivity. We observe once again that varying the number of permutations ($N_{\text{perms}} = 1$ vs. $N_{\text{perms}} = 100$) and the $K$-factor plays a critical role in stability. In the ♖ and ♗ scenarios, with $N_{\text{perms}} = 100$ and $K = 1$, we notice discrepancies in the models' rankings. This contrasts with $K = 16$, where rankings are more consistent and accurate. The slower updates from $K = 1$ suggest this setting may be too conservative to capture transitive relations quickly, leading to inconsistencies.

## 6 Validation on Real-World Human Feedback

Building on the insights gained from synthetic data experiments, this section extends the validation of the Elo rating system to real-world human feedback. Our objectives are twofold: first, to ascertain how the properties demonstrated using synthetic data generalize to real human annotations, and second, to evaluate the Elo rating system's utility for assessing LLMs in practical settings.

**Experimental Setup** We use the *LMSYS - Chatbot Arena* dataset [34], an open-source collection of human preference data derived from unique users' interactions with two distinct models responding to a set of user-defined prompts. To align with our methodology from synthetic data analysis, tie outcomes have been excluded from this analysis to focus specifically on the implications of win-loss dynamics. We select pairs of models (A vs. B) from the initial dataset that feature at least 300 non-tie comparisons. This threshold ensures statistical robustness and allows us to include cases where win rates are closely contested, which can lead to more sensitive ratings. These pairs predominantly involve models from the GPT-4 family [42] and the Claude family [1]. A comprehensive list of model pairs is included in Appendix C under Table 3, and a subset discussed here is shown in Table 2. The recorded win rates primarily exhibit skewed preferences, with the

Table 2: Win rates per evaluated model across selected paired comparison experiments.

| Experiment | Win Rates |
|---|---|
| GPT-4-0314 | 0.51 |
| GPT-4-0613 | 0.49 |
| Claude-1 | 0.59 |
| Claude-2.1 | 0.41 |
| GPT-4-0314 | 0.65 |
| Claude-2.1 | 0.35 |
| GPT-4-0613 | 0.61 |
| Claude-2.1 | 0.39 |
| GPT-4-1106-preview | 0.86 |
| GPT-4-0613 | 0.33 |

exception of the GPT-4-0314 vs. GPT-4-0613 pairing, indicating comparable performance levels (see Table 2). Given the variable number of evaluations per pair in the original dataset, we standardize this by sampling a fixed number, $N_{\text{sample}}$, for each pair to align with the controlled conditions used in synthetic analyses. When sampling to $N_{\text{sample}}$, we ensure that the resulting win rates accurately represent the original dataset's findings, providing a faithful evaluation of recorded model performance. This standardization facilitates a more reliable comparison and assessment of the Elo rating

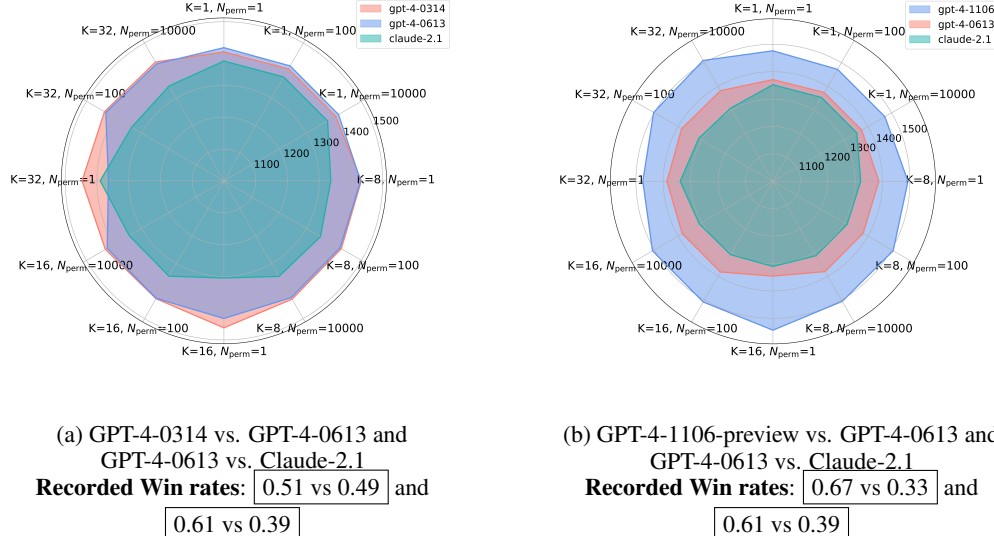

(a) GPT-4-0314 vs. GPT-4-0613 and
GPT-4-0613 vs. Claude-2.1
**Recorded Win rates**: $\boxed{0.51 \text{ vs } 0.49}$ and
$\boxed{0.61 \text{ vs } 0.39}$

(b) GPT-4-1106-preview vs. GPT-4-0613 and
GPT-4-0613 vs. Claude-2.1
**Recorded Win rates**: $\boxed{0.67 \text{ vs } 0.33}$ and
$\boxed{0.61 \text{ vs } 0.39}$

Figure 5: Elo scores ($S_A$, $S_B$ and $S_C$) for three models at different configurations of $N_{perms} = \{1, 100, 10000\}$ and $K$-factor = $\{1, 8, 16, 32\}$. The intersections of score lines in 5a indicate fluctuating relative rankings, highlighting inconsistency especially pronounced among models with close performance levels. In contrast, 5b demonstrates more stable relative rankings in conditions where win rates are more skewed.

system under real-world conditions. In line with our previous analyses, we continue to explore the influence of variations in $N_{\text{perms}} = \{1, 100, 10000\}$ and the $K$-factor (ranging from 1 to 36) on Elo score robustness and reliability. We examine scenarios where one model decisively outperforms another (e.g., Claude-1 vs. Claude-2.1) and cases where models are nearly evenly matched (e.g., GPT-4-0314 vs. GPT-4-0613).

**Key Findings** Our analysis of real-world human feedback data confirms that the stability of Elo ratings is influenced by disparities in win rates, analogous to win probabilities in synthetic data, and by the choice of hyperparameters $K$-factor and $N_{\text{perms}}$. In cases where the models show a clear difference in performance as indicated by their win rates, such as in the Claude-1 vs. Claude-2.1 experiment, Elo ratings remain notably consistent across different $K$-factors and $N_{\text{perms}}$ configurations (see Figure 4a). On the other hand, in cases like the GPT-4-0314 vs. GPT-4-0613 experiment where win rates are closely matched, the Elo rating system exhibits higher volatility at $N_{\text{perms}} = 1$ but gains stability with larger $N_{\text{perms}}$ settings (100 and 10000), especially at lower $K$-factors (see Figure 4b). The magnitude of Elo score differences in these experiments illustrates that larger $K$-factor and $N_{\text{perms}}$ values can amplify or reduce the perceived performance gap between models, reflecting the critical role of these parameters in evaluation sensitivity.

Regarding the conservation of transitivity, our findings indicate that this property is not universally maintained across real-world human evaluations and synthetic scenarios (see Section 5). The relative rankings of models with similar performance levels are particularly sensitive to the choice of hyperparameters. Consequently, one should exercise caution in drawing conclusions from the Elo scores, especially in the absence of extensive paired comparison data as required by the combination formula 5. These observations are consistent with the trends from our synthetic data experiments.

## 7 Related Work

Several works have proposed improvements to the Elo rating system. Variants such as Glicko [18, 19, 20] and TrueSkill™ [24, 39] have incorporated more complex statistical methods into the original Elo framework, to address some of the limitations of the Elo rating system, particularly in the context of games with more than two players or teams, or games with more complex outcomes

than just win or loss. There is also ongoing research into the efficacy of these systems in diverse and dynamic environments [11, 7]. Prior work has demonstrated some limitations of Elo in maintaining transitivity, especially in non-transitive cyclic games such as rock-paper-scissors and StarCraft II [7, 52]. However, our work diverges by focusing on the reliability of Elo applied to large language model systems. To date, there has not been a comprehensive evaluation in this context.

Independent from Elo, numerous studies have explored how sensitivity to hyperparameters can undermine the generalization of findings [41, 36, 23, 27, 9] in machine learning. This forms part of a wider body of work that considers which factors influence reliability and reproducibility [21, 22, 5, 14]. Notable directions includes studies on the impact of random seeds [40, 37, 51], model design choices [46, 48, 43, 28, 47], the use of data parallelism [45], hardware [58] and test set construction [49, 30, 38]. Our work is complementary to these efforts, providing a rigorous evaluation of the impact of key hyperparameters and experimental settings on Elo performance.

## 8    Empirical Guidelines for Robust Elo-based Evaluation of LLMs

In this section, we distill essential practices for enhancing the reliability of Elo-based evaluation of language models. These guidelines, derived from our empirical findings, differ notably from some conventional Elo settings and have significant implications for current real-world applications:

- **Achieving Score Stability**: To obtain stable and reliable Elo ratings, it's recommended to run numerous permutations, ideally with $N_{\text{perms}} \geq 100$. This approach significantly improves the consistency of outcomes over single or fewer permutations commonly used.
- **Adjusting the $K$-factor**: A smaller K-factor may reduce significant rating fluctuations when models have closely matched win rates.
- **Rapid Convergence for Clear Winners**: When there is a clear performance disparity between models, a higher K-factor accelerates the alignment of Elo ratings with the models' "true" performance levels. This is in stark contrast to traditional uses of Elo ratings, where a one-size-fits-all K-factor is frequently applied.
- **Transitivity is not guaranteed**: The assumption that ($A$ beats $B$ and $B$ beats $C$ implies $A > C$) is not consistently valid in Elo ratings. This is particularly invalid when models have similar performance levels, challenging a common assumption in many Elo-based evaluations.

These guidelines serve as empirically grounded recommendations to improve the robustness and interpretability of Elo-based evaluations for LLMs. Following these best practices will help in yielding more reliable conclusions on models' performance via human judgment.

## 9    Conclusion and Limitations

This paper presents a comprehensive study on the reliability of the Elo rating system for evaluating LLMs through human feedback within an axiomatic framework. We identify various factors that influence the robustness of Elo ratings and provide guidelines for their effective application in real-world scenarios. While our findings establish an essential foundation, they are by no means exhaustive. Future work could extend the present study by considering tie outcomes and adopting multi-category Bernoulli synthetic data to more closely simulate the varied landscape of human feedback. Such extensions could yield additional insights into the convergence properties of the Elo rating system in the fast-evolving field of language models.

## 10    Impact Statement

The implications of our work are significant in fields relying on LLMs for decision-making, content generation, and more. Improving the evaluation methods of LLMs contributes to the development of AI systems that are more reliable and trustworthy. This research also holds the potential to influence evaluation practices in other sectors that employ the Elo rating system, broadening its relevance and utility. However, it also emphasizes the need for cautious, informed application of Elo ratings to prevent misinterpretation or reliance on Elo-based rankings, particularly when the performance of models is comparable. As LLMs become more integrated into societal frameworks, ensuring the robustness and reliability of their evaluation mechanisms is paramount to fostering ethical, beneficial AI advancements.

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

## A  Extension to Multiple Outcomes

For scenarios where outcomes can extend beyond wins and losses, such as a tie option, one could make use of the multinomial distribution for the synthetic data generation process. For the three outcomes; win, loss, and tie, one sample according to the distribution:

$$P\left(n_{\text{win}}, n_{\text{loss}}, n_{\text{tie}}; N, p_{\text{win}}, p_{\text{loss}}, p_{\text{tie}}\right)$$
$$= \frac{N!}{n_{\text{win}}! n_{\text{loss}}! n_{\text{tie}}!} p_{\text{win}}^{n_{\text{win}}} p_{\text{loss}}^{n_{\text{loss}}} p_{\text{tie}}^{n_{\text{tie}}} \tag{8}$$

## B  Impact of Ordering on Elo Ratings: Skewed Win Rates

We summarize our findings on the impact of match sequences on Elo ratings for winning probabilities $Prob(A \text{ beats } B) \geq 0.65$.

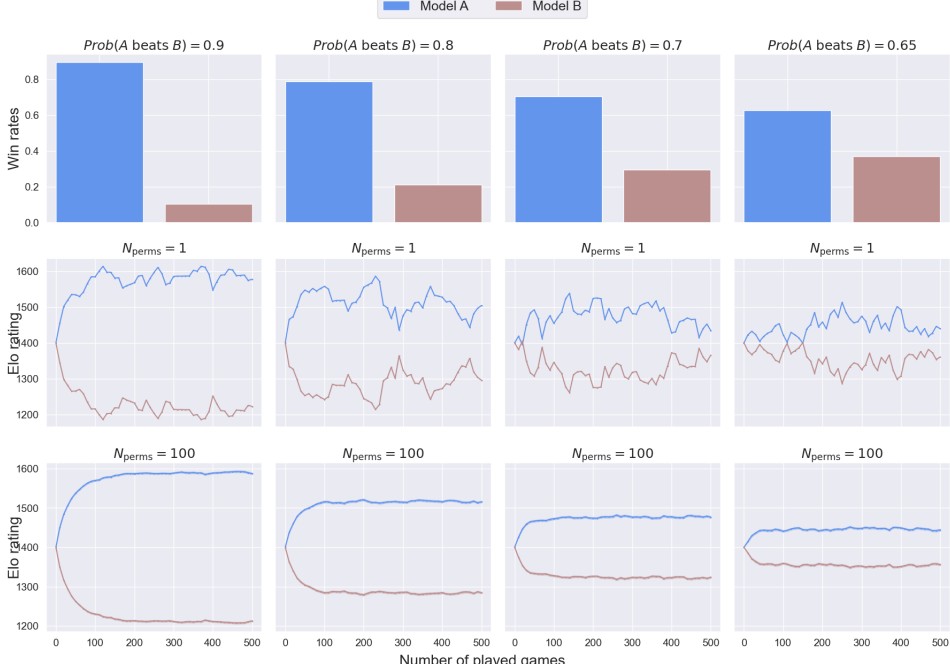

Figure 6: **Impact of win probabilities and permutation sampling on Elo ratings**: Comparing Model A and Model B across three different win probabilities ($Prob(A \text{ beats } B) = 0.9, 0.8, 0.7, 0.65$) with two levels of permutation sampling ($N_{\text{perms}} = 1$ and $N_{\text{perms}} = 100$). The top row displays the observed win rates, the middle row illustrates Elo ratings with a single permutation, and the bottom row shows the mean and standard error of the mean (SEM) of Elo ratings across 100 permutations.

## C  Chatbot Arena Human Preference Data Preparation

For the experimental validation of the Elo rating system using real-world data, we utilize the LMSYS dataset from [34]. We first vizualize the first 100 unique paired comparisons sorted in descending order by the number of recorded evaluations. The distribution of tie vs. non-tie outcomes is shown in Figure 7. To refine the dataset for our analysis, we exclude tie results, focusing exclusively on win-loss dynamics. The remaining dataset is further filtered to identify pairs with at least 300 non-tie comparisons. This threshold of 300 allows us to encompass a broad spectrum of comparison scenarios, ranging from skewed win rates ($\text{Model}_A \gg \text{Model}_B$) to closely matched ($\text{Model}_A \simeq \text{Model}_B$). These selected paired comparisons are depicted in Figure 8.

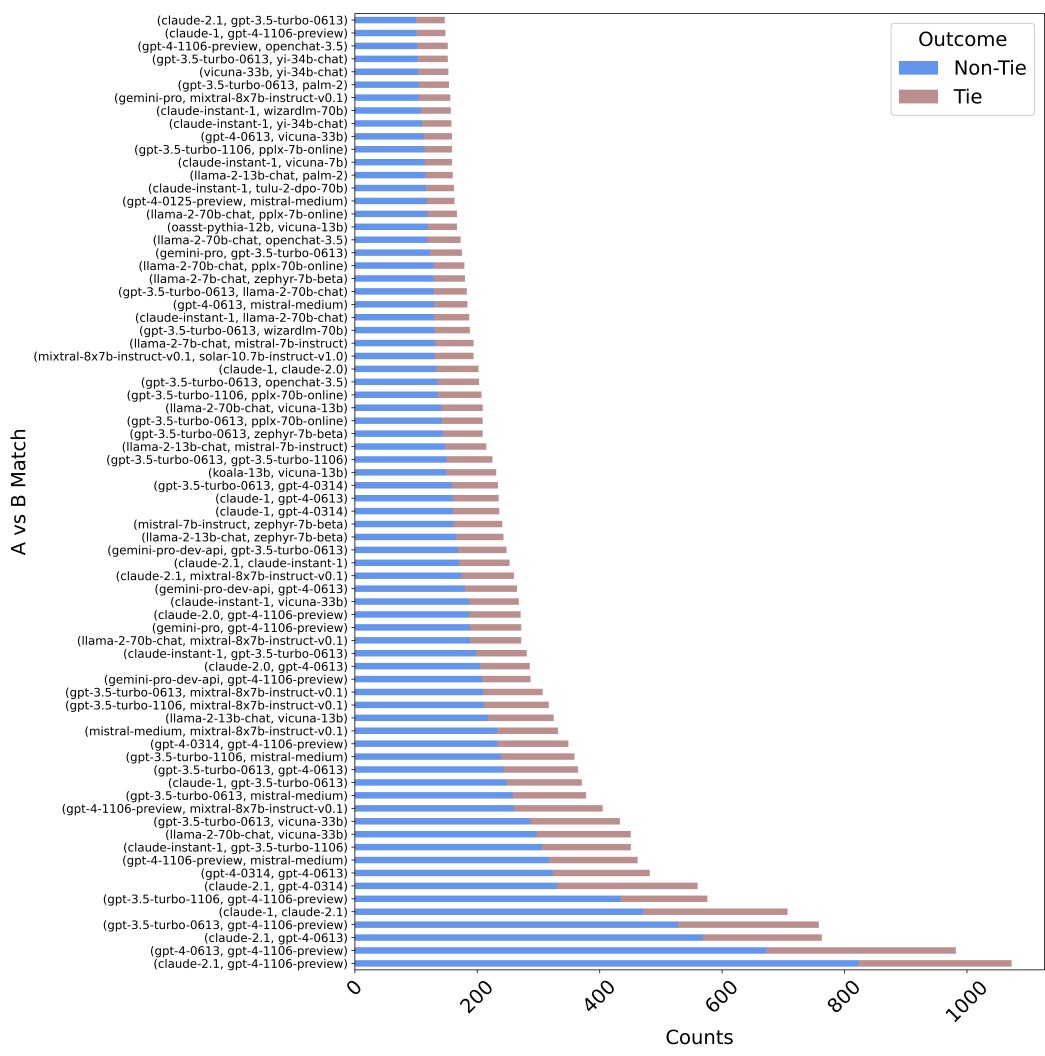

Figure 7: Initial Distribution of Tie vs. Non-Tie Outcomes: A visual overview of the first 100 paired comparisons from the LMSYS dataset ordered by evaluation sizwe

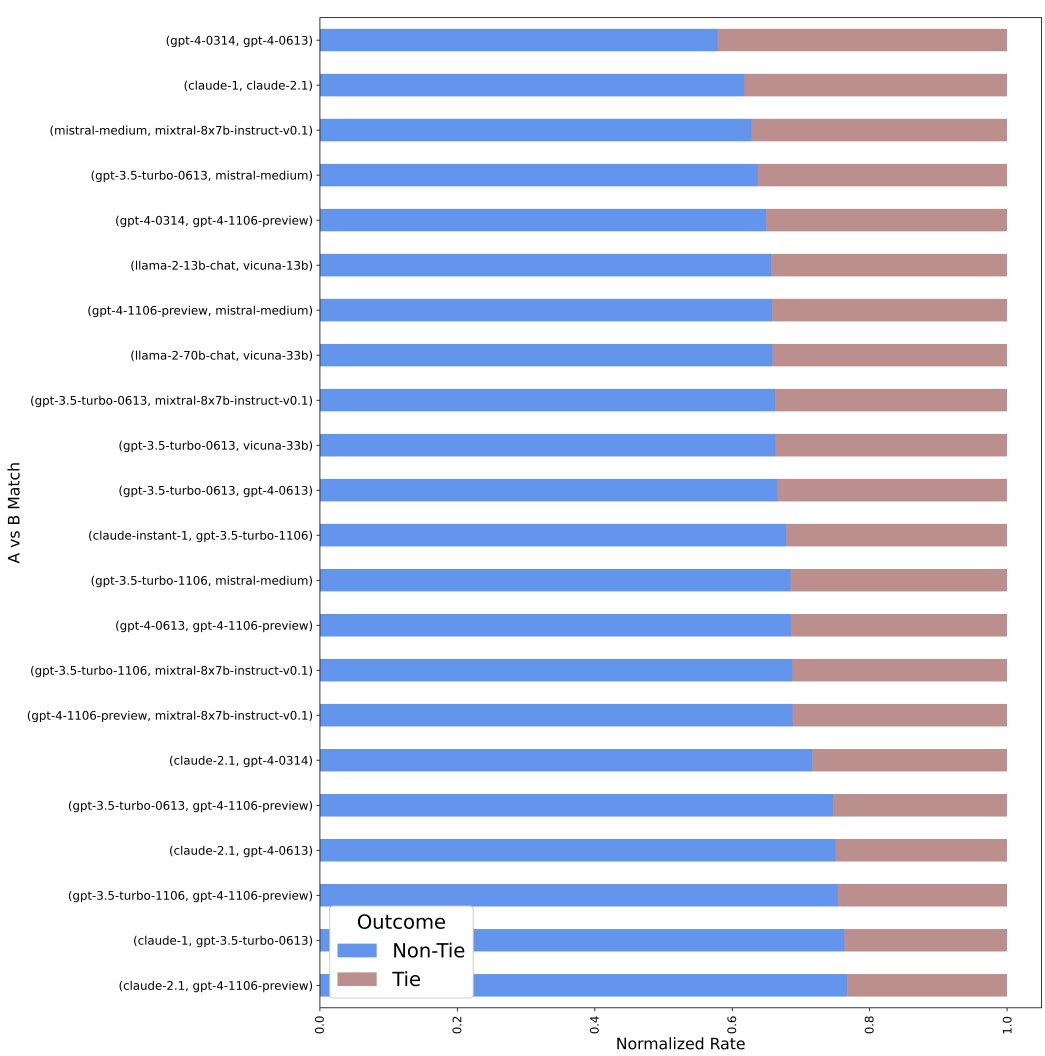

Figure 8: Normalized Tie vs. Non-Tie outcomes for A vs B model comparisons with at least 300 evaluations.

Table 3: Win Rates for Paired Model Evaluations: This table presents the initial match counts and win rates for model comparisons from [34] where each pair has at least 300 matches, excluding ties. Results from a fixed sample size of 300 are shown to demonstrate model performance under controlled sampling conditions.

| Experiment | Original Size | Win Rates (%) | Sample Size | Sampled Win Rates (%) |
|---|---|---|---|---|
| gpt-4-0314
gpt-4-0613 | 324 | 50.93
49.07 | 300 | 51.00
49.00 |
| gpt-4-1106-preview
gpt-4-0613 | 673 | 67.01
32.99 | 300 | 67.00
33.00 |
| gpt-4-1106-preview
gpt-3.5-turbo-0613 | 528 | 81.25
18.75 | 300 | 81.33
18.67 |
| gpt-4-1106-preview
gpt-3.5-turbo-1106 | 434 | 86.18
13.82 | 300 | 86.33
13.67 |
| claude-1
claude-2.1 | 471 | 58.60
41.40 | 300 | 58.67
41.33 |
| gpt-4-0314
claude-2.1 | 331 | 65.26
34.74 | 300 | 65.33
34.67 |
| gpt-4-0613
claude-2.1 | 569 | 61.16
38.84 | 300 | 61.00
39.00 |
| gpt-4-1106-preview
claude-2.1 | 823 | 75.21
24.79 | 300 | 75.33
24.67 |
| claude-instant-1
gpt-3.5-turbo-1106 | 306 | 56.86
43.14 | 300 | 57.00
43.00 |
| gpt-4-1106-preview
mistral-medium | 317 | 71.92
28.08 | 300 | 72.00
28.00 |

