# OpenReview forum: "Elo Uncovered: Robustness and Best Practices in Language Model Evaluation"
_NeurIPS.cc/2024/Conference — NeurIPS 2024 poster_

### Official Review · Reviewer_hqxB · 2024-06-18

**Soundness:** 3
**Presentation:** 2
**Contribution:** 3
**Rating:** 7
**Confidence:** 5

**Summary:**

This paper conducts an empirical study on the reliability and transitivity of the Elo rating system when evaluating Large Language Models. By conducting experiments on both simulated data and real-world scenarios, this paper suggests that certain parameter setting guidelines should be followed to ensure the stability of evaluation results when applying the Elo rating system to LLM assessments. The paper provides insights and guidance for future research and applications of the Elo rating system in various LLM evaluation contexts.

**Strengths:**

- This paper provides a detailed experimental analysis of the practicality of the currently common Elo rating system used for evaluating the performance of Large Language Models (LLMs).

- Through appropriate experimental setups, this paper explores the impact of various factors on the Elo rating system results, including the number of sequence arrangements, the number of games, the K value, and win rates.

- By combining results from simulated experiments and real-world scenarios, this paper demonstrates good reliability in its findings.

**Weaknesses:**

- This paper explores the experimental results with up to three players. It would be more meaningful if the conclusions could be further generalized to settings with more players.

- In reality, $N_{perms}$ can be set very large without obvious computational cost to stable the evaluation results. Therefore, in multiple experiments, I would prefer to see results with settings like $N_{perms}$ = 10k, similar to those in Figure 2.

**Questions:**

My main concerns are listed in Weaknesses

**Limitations:**

I think that this paper has no significant limitations apart from what is mentioned by the authors in Section 8.

---

> ### Author Rebuttal · Authors · 2024-08-06
>
> We thank Reviewer [ hqxB] for their positive feedback including observing our **“detailed”** and **“appropriate”** experiment setup as well as noting the breadth of ablation including **“sequence arrangements, the number of games, the K value, and win rates”**. We appreciate the comment that the rigor of the analysis provides **“good reliability”** in the findings. We take this opportunity to address R hqxB helpful feedback and questions below:
>
> > This paper explores the experimental results with up to three players. It would be more meaningful if the conclusions could be further generalized to settings with more players.
>
> We thank the reviewer for this suggestion. Introducing each new model necessitates a comprehensive comparison with all previous models, which can quickly become overwhelming. To provide an overview of the landscape, we targeted a wide range of models with different architectures (encoder-decoder and decoder-only), training data (public, private), and model sizes (3B to 100s of billions). Additionally, our simulated experiments are devoid of any model artifacts and potential biases. This approach helps contextualize and establish our findings within the real world.
> Lastly, in the shared response above, we added analysis of a recently released dataset including 20 models (such as GPT-4, Claude-v1, and Palm-2) and observed confirmation of our findings in the manuscript. We will update the final manuscript with these results.
>
> > In reality, Nperms can be set very large without obvious computational cost to stable the evaluation results. Therefore, in multiple experiments, I would prefer to see results with settings like Nperms = 10k, similar to those in Figure 2.
>
> We agree with the reviewer that the cost of large Nperm is offline and we can increase this number to an extent. We compute and update Elo ratings for "$N_{\text{pairwise games}} \times N_{perm}$" instances. As the Number of pairwise games ($N_{\text{pairwise games}}$) grows by increasing the number of models and the number of prompts in the dataset, running a high number of permutations will slow down the evaluation process. More importantly, our study shows that we do not need to run unnecessarily high numbers of permutations. As shown in Figure 2 in the paper, we observe stable results with $N_{perm} = 100$ with minuscule standard errors of the mean and conclude that there is no need to run a higher number of permutations. With the new Chatbot Arena dataset added in the shared response section of the rebuttal, we have 33k prompts and observed stable results with $N_{perm} = 1000$.
>
> ---
>
> We hope that we were able to clarify the reviewer’s questions. If so, we ask that you consider raising your score to reflect. If you have any further questions or concerns during the rebuttal period, please let us know. We are happy to provide any necessary clarifications.

---

> > ### Author Response · Authors · 2024-08-08
> >
> > Now that the discussion is underway, we wanted to ask Reviewer **hqxB** if there are any follow-up points we can clarify. If there are no further points of clarification regarding the manuscript and using a dataset with more players (with the new experiments shared here: https://openreview.net/forum?id=Pc9LLjTL5f&noteId=Kv4lOJURpk) and the need to run high number of permutations, we would ask that reviewer Reviewer **hqxB** consider increasing their score to reflect the additional experiments, improvements and clarifications we have provided. We are very happy to continue to engage and answer any questions.

---

> > ### Comment · Reviewer_hqxB · 2024-08-12
> >
> > I appreciate the author's detailed response, which has addressed my concerns to some extent.

---

### Official Review · Reviewer_9SNU · 2024-07-12

**Soundness:** 2
**Presentation:** 1
**Contribution:** 3
**Rating:** 4
**Confidence:** 4

**Summary:**

Updated after response: I think this paper could provide some valuable contributions to the community, but has real presentational issues. The authors fixed some of them that had an easy fix  (which I pointed out) during the rebuttal. However, I believe the presentational standards (and therefore the understandabiluty of the contribution) are not up to par for this conference, and I thus stand by my borderline reject recommendation.

This paper studies the Elo rating system, recently popular in the context of ranking LLMs. Specifically they study two axioms that evaluation methods should adhere to: reliability and transitivity. They find that Elo ratings for LLMs are highly sensitive to the order of comparisons and the choice of hyperparameters, and that desirable properties such as transitivity are not always guaranteed.

In their first experiment, the authors focus on order sensitivity. They conduct a controlled experiment with synthetic human feedback, in which they compare various model pairs with different probabilities of winning a pairwise comparison.

They show that, with only order of comparisans (which, if I understand correctly, would be the actual scenario in which we deploy this?), if model A has a probability 40<P<60 to win  a pairwise comparison with model B, even 500 games are not enough to converge to the conclusion that model A is better than model B. With 100 orders, the difference is clear after few games, though (semi) convergence happens only after around 80.  For smaller differences (51%), even 100 orders do not reliably show the difference between models. For differences of P(a_win > 60) Elo results are stable (?) (I can't fully match this statement in the text with the figures).

Next, the authors investigate the impact of K-factor (which, I learned from Wikipedia, is the maximum number of points a ranking can be updated after a match). They show that higher K-factors eacerbate instability for single orderings for models that have similar performances, but that faster convergence is observed for higher K-factors when models diverge more.

Then, the authors investigate transitivity, using a set of three models with different respective win probabilities. They find that when there is a clear disparity between models, Elo ratings accurately reflect the expected hierarchy, but in more complex cases the rankings become less stable, challenging the assumption of transitivity. K-factor and number of orderings play a key roal in the stability of the rankings.

The authors confirm several of their findings using real human feedback data. And wrap-up their paper with a list of recommendations for using Elo ratings.

**Strengths:**

The topic studied in this paper (Elo ratings for LLMs) is important, and the experiments are extensive. The paper provides several interesting lessons for using Elo ratings for LLMs that I believe can (positively) influence how we compare models in rankings based on pair-wise comparisons.

**Weaknesses:**

While I think this paper could potentially be very impactful and has many interesting results, in its current state it does not seem ready for publication. It took me excessively long to review, because the presentation is rather messy, and not clear at several points.  Some of those things are easily fixable, e.g.:
- The Elo rating system is briefly explained in an appendix, but given the importance of the concept it should be in the main text (if space is needed, I think equation 5 describing transitivity can safely be moved to the appendix).
- Some explanation of the K-factor should also be included. It is sort of described, but not really explained or defined, not even in the appendix as far as I can see).
- Maybe I missed it, but I don't think it is listed how Elo ratings are most commonly used for LLMs, is it with only a single ordering?
- Table 1 is somewhat difficult to read, why not make a plot similar to Figure 1 but with multiple lines for the different models?
- In the introduction -- impliations of our work -- it is listed that the work is particularly important in situations where model performances are closely matched, which is a common occurrence in many real-world scenarios, whereas in 4.2 it is listed that the result is relevant because skewed win-rates are common in real-world scenarios (or maybe that is not what is meant, if not I am not sure why it is mentioned that there are skewed win-rates in real world scenarios)

More problematic are the unclarities concerning the results, for example:
- It is not clear with what K-factor the main experiment (4.1) is conducted (or at least I couldn't find it)
- in 4.1 it is stated that for P(A beats B) >= 0.6 Elo results are stable; I can't match this result with Figure 1 (middle column), where there seems to be quite some variation for P(a beats B) = 0.55 as well as P(a beats B) = 0.6
- The idea of comparing a single sequence with 100 different orderings is generally strong, but doing only one single sequence and one series of permutiations is not very informative in this case: what we actually want to know is how often a single sequence or an average over 100 sequences would give us the right results. Perhaps this single sequence was a very unlucky one among many that give us the correct result, or the 100 permutations were instead 100 lucky permutations. To really make the point, we'd need the variation between single sequences, and multiple selections of 100 permutations. (similar for Figure 4)
- In 4.2 it is listed that faster convergence is observed for higher K-factors, but I don't understand the evidence for that. I don't think this is visible from Figure 3, and I couldn't find an appendix plot either.

**Questions:**

- If I understand correctly, the number of permutations is the number of game orders, why not use a word that matches that closely, e.g. 'game orders' or 'orderings'?

**Limitations:**

Yes

---

> ### Author Rebuttal · Authors · 2024-08-06
>
> We thank Reviewer [9SNU] for their helpful feedback and their observations noting that our study is **“important”** and the experiments are **“extensive”**. We also hugely appreciate the reviewer’s note that this paper provides **“several interesting lessons for using Elo ratings for LLMs”** that **“can (positively) influence how we compare models”**. We take this opportunity to address R 9SNU helpful feedback and questions below:
>
> > Table 1 is somewhat difficult to read, why not make a plot similar to Figure 1 but with multiple lines for the different models?
>
> We thank the reviewer for the helpful feedback to make this table more readable and understandable. We will take the great suggestion from this reviewer into account to update the manuscript. To clarify, the main takeaway from Table 1 is ​​the red star values for which we observe the cases where the Elo score fails to accurately reflect the expected hierarchy of models.
>
> > In the introduction -- implications of our work -- it is listed that the work is particularly important in situations where model performances are closely matched, which is a common occurrence in many real-world scenarios, whereas in 4.2 it is listed that the result is relevant because skewed win-rates are common in real-world scenarios (or maybe that is not what is meant, if not I am not sure why it is mentioned that there are skewed win-rates in real world scenarios)
>
> Thanks for pointing out this unclarity in our manuscript. We believe that both scenarios (closely matched models and skewedly performing models) are equally important and they both occur in real-world model comparisons as is evident on the Chatbot Arena ranking. Our paramount goal in this work is to study the reliability and robustness of comparison methods such as Elo across all scenarios. We will clarify this detail in both the introduction and section 4.2.
>
> > It is not clear with what K-factor the main experiment (4.1) is conducted (or at least I couldn't find it)
>
> Thanks for pointing out this missing point. We use $K=16$ for all experiments where we don’t explore different values of $K$. We will make this clear in the manuscript.
>
> > It is not clear with what K-factor the main experiment (4.1) is conducted (or at least I couldn't find it) in 4.1 it is stated that for P(A beats B) >= 0.6 Elo results are stable; I can't match this result with Figure 1 (middle column), where there seems to be quite some variation for P(a beats B) = 0.55 as well as P(a beats B) = 0.6*
>
> We thank the reviewer for giving us the opportunity to clarify the results presented in Section 4.1.
>
> - $P(A \text{ beats } B) > 0.6$: The Elo results for models where the win probability is $P(A \text{ beats } B) > 0.6$ are presented in Figure 6 (and not Figure 1). There we observe that with both one sequence and 100 permutations (row 2 and row 3 respectively in Figure 6), the Elo convergence is stable, i.e., the rankings of the models don’t change over the number of games played.
>
> - $P(A \text{ beats } B) = 0.6$ and $P(A \text{ beats } B) = 0.55$: In Figure 1 we observe a slight instability with one one sequence but no change of model orders with increasing the number of games ($N_{perm} = 100$): model A achieves a higher Elo score with the first few games and stays on top as we increase the number of games.
>
> - $P(A \text{ beats } B) = 0.51$: we observe a different pattern where depending on how many games we conduct the model order changes: at 100 model B is better than A and at 250 model A is better than model A.
> We make sure that the Figures 1 and 6 are consolidated and the description of the figures are more clear in the final manuscript.
>
> > The idea of comparing a single sequence with 100 different orderings is generally strong, but doing only one single sequence and one series of permutations is not very informative in this [...]
>
> Reviewer [9SNU] raised a valid concern about the potential bias from comparing one sequence with $N$ permutations, as the single sequence could be unusually lucky or unlucky. To address this, we conducted new experiments using a dataset with 33k prompts, as detailed in the shared response section above.
>
> > In 4.2 it is listed that faster convergence is observed for higher K-factors, but I don't understand the evidence for that. I don't think this is visible from Figure 3, and I couldn't find an appendix plot either.
>
> We would like to thank the reviewer for pointing out this vagueness in Figure 3. We will clarify the convergence details in this figure by adding the time dimension and showcase the speed of convergence more clearly.
> To explain the current figure we don’t currently show the traditional notion of convergence, but rather show stability. Here we present the final Elo score differences $(SA − SB)$ for different parameter values (different $K$-factors and win probability scenarios). For all matches model A beats model B (albeit with different probabilities of win) and a positive value for this difference is desired. For larger $K$ values, we observe larger score differences $(SA − SB)$ indicating that the system is more stable and confident in predicting model A is better than model B.
> Nonetheless, we will ensure that the final manuscript is clear on this detail.
>
> > If I understand correctly, the number of permutations is the number of game orders, why not use a word that matches that closely, e.g. 'game orders' or 'orderings'?
>
> That is true. We mainly use permutation because of its convention of use as a mathematical definition to describe a number of orderings and common use in combinatorics. We will use “game orders” and will define this term more explicitly and clearly in the final manuscript.
>
>
> ---
>
> We hope that we were able to clarify the reviewer’s questions and concerns. If so, we ask that you consider raising your score to reflect. If you have any further questions or concerns during the rebuttal period, please let us know. We are happy to provide any necessary clarifications.

---

> > ### Author Response · Authors · 2024-08-08
> >
> > Now that the discussion is underway, we wanted to ask Reviewer **9SNU** if there are any follow-up points we can clarify. If there are no further points of clarification regarding the manuscript and using a larger dataset to avoid bias (with the new experiments shared here: https://openreview.net/forum?id=Pc9LLjTL5f&noteId=Kv4lOJURpk), clarification on tables in question as well as missing parameter values, and explanation of a figure in question, we would ask that reviewer Reviewer **9SNU** consider increasing their score to reflect the additional experiments, improvements and clarifications we have provided. We are very happy to continue to engage and answer any questions.

---

> > > ### Comment · Reviewer_9SNU · 2024-08-10
> > >
> > > Thank you, I appreciate the responses and comments. Content-wise I think your paper could be very influential. As I said in the review, however, I found the paper very hard to read. While I really appreciate that the authors have fixed some of the 'easy' fixes I pointed out, I think that this paper would benefit from a more thorough haul-over and proofreading, and think it would be a waste to get it out with a suboptimal presentation. I therefore feel that it should under go another round of review before being published, so I do not wish to update my score.

---

> > > > ### Author Response · Authors · 2024-08-10
> > > >
> > > > We thank the reviewer for their thoughtful feedback and for acknowledging the  scientfiic merit of our contributions,  "**Content-wise I think your paper could be very influential**". We understand the concerns about readability, but we respectfully believe this **should not** be the sole reason for withholding support for acceptance. We would like to kindly refer to the NeurIPS reviewer guidelines, which suggest:
> > > >
> > > > > **Incorrect claims or methodology are the primary reason for rejection**
> > > >
> > > > and
> > > >
> > > > > **The “Overall Score” for each submission should reflect your assessment of the submission’s contributions**
> > > >
> > > > We are committed to addressing any readability issues in the revision process, and we believe the concerns raised about readability (*the concept of elo should be introduced in main text, k-factor should be better described, table 1 could be visualized better*) are very addressable especially given an additional page will be made available to camera-ready papers which are accepted (some of these readability issues stem from the page limit which forced some details to be moved to appendix).
> > > > We thank the reviewer since their feedback has already greatly improved the presentation of our work, and hope they will reconsider their score. However, regardless we appreciate the reviewer for engaging fully during this discussion period.

---

### Official Review · Reviewer_NZSD · 2024-07-12

**Soundness:** 3
**Presentation:** 3
**Contribution:** 3
**Rating:** 5
**Confidence:** 2

**Summary:**

Evaluation plays an important role in LLM research. Previous works have assessed the performance using the Elo rating system, which is designed for ranking players in games. This paper shows that Elo rating does not always satisfy two critical axioms, reliability and transitivity. So the rankings of the models are not accurate. Transitivity is whether a consistently higher-rated model outperforms those with lower ratings in all scenarios. Reliability has two aspects, sensitivity to ordering and sensitivity to hyperparameters, how the Elo scores are affected by the order in which models are compared and the sensitivity to hyperparameter adjustments, particularly the K-factor.

The authors first validate the properties of Elo using synthetic data generation via Bernoulli processes. Then extend the validation to include real-world human feedback. The findings are that the stability of Elo ratings is influenced by disparities in win rates, analogous to win probabilities in synthetic data, and by the choice of hyperparameters K-factor and Nperms. Finally the authors provide empirical guidelines for robust Elo-based Evaluation.

**Strengths:**

1. The evaluation of LLMs plays an important role in LLM research. The reliability of the Elo rating is an interesting and meaningful research topic.

2. The authors highlight two Axioms for Elo-based evaluation, which is reasonable and useful.

3. This paper verifies the findings from both synthetic human feedback and real-world human feedback.

4. This paper provides several empirical guidelines for robust elo-based evaluation.

**Weaknesses:**

1. The writing is not very friendly to the reader unfamiliar with the Elo rating.

2. This paper only conducts experiments on three LLMs.

**Questions:**

1. In line 93 you mention the K factor, but it is hard to understand for the reader unfamiliar with the Elo rating.

2. How can the Elo rating be generalized to multiple-task evaluation?

3. Why not conduct experiments on more widely used API-based LLMs, such as ChatGPT, Gemini, and Claude-3?

**Limitations:**

This paper has no potential negative societal impact of their work.

---

> ### Author Rebuttal · Authors · 2024-08-06
>
> We thank Reviewer [ NZSD] for their helpful feedback and observations, noting that our study is **“an interesting”** and **“meaningful research topic”** and find our axiom definitions **“reasonable and useful”**. We take this opportunity to address R NZSD concerns and feedback below.
>
> > The writing is not very friendly to the reader unfamiliar with the Elo rating. In line 93 you mention the K factor, but it is hard to understand for the reader unfamiliar with the Elo rating.
>
> Thank you for your feedback. We agree that this could be more kind to the reader, we will provide a more in-depth introduction to the Elo rating system in the final manuscript, including the impact of the $K$-factor on it.
>
> > How can the Elo rating be generalized to multiple-task evaluation?
>
> This is indeed an exciting question. One method would be to just run experiments per task and obtain an “Elo per task" rating and then average these ratings across different tasks. We want to note that even the single task setup currently used with LLMs is complex enough since the "game" (prompt) changes every round and the win condition is subjective (due to human judgment). However adding a multi-task angle to capture the complexity of various domains and human feedback would be interesting.
> Lastly, there are more sophisticated rating methods such as Trueskill that can be useful for such scenarios. TrueSkill [1] is a Bayesian ranking system that can handle multiple tasks using a vector of skill parameters, each representing a different task.
>
> > Why not conduct experiments on more widely used API-based LLMs, such as ChatGPT, Gemini, and Claude-3?
>
> We thank the reviewer for this question. To start, we want to explain how the core of our findings are model-agnostic. Relative differences and win rates are all that matter when we are studying the dynamics of Elo. With introducing more powerful models, the results could be reduced to one of the two cases we already have: model pairs that are close in performance or model pairs that have a large performance gap. The only new dynamics that may occur with many SOTA models is saturating Elo scores at the top.
> But that isn't very relevant here since it's easy to still find prompts that all LLMs are not good at so we haven't found a saturation point yet.
> Introducing each new model necessitates a comprehensive comparison with all previous models, which can quickly become overwhelming. The primary reason for not using API-based LLMs is their potential for behind-the-scenes updates that can alter performance over the evaluation time and impact replicability [2]. Hence, for this treatment, we restricted consideration to open weights which were highly performant and we have control over when collecting preference feedback over a period of time.
> Lastly, to cover a larger model list we looked at the Chatbot Arena dataset with 20 models and present the findings in the shared response of this rebuttal. We confirm our previous findings with this dataset.
>
>
> [1] Ralf Herbrich, Tom Minka, and Thore Graepel. 2006. TrueSkill™: a Bayesian skill rating system. In Proceedings of the 19th International Conference on Neural Information Processing Systems (NIPS'06). MIT Press, Cambridge, MA, USA, 569–576.
>
> [2] Luiza Pozzobon, Beyza Ermis, Patrick Lewis, Sara Hooker. 2023. On the Challenges of Using Black-Box APIs for Toxicity Evaluation in Research
>
> ---
>
> We hope that we were able to satisfy the reviewer’s asks by running additional experiments on a much larger dataset and clarifying the details in question. If so, we ask that you consider raising your score to reflect and please do let us know if anything requires further clarification.

---

> > ### Author Response · Authors · 2024-08-08
> >
> > Now that the discussion is underway, we wanted to ask Reviewer **NZSD** if there are any follow-up points we can clarify. If there are no further points of clarification regarding the manuscript and covering more widely used models such as claude and gpt4 (with the new experiments shared here: https://openreview.net/forum?id=Pc9LLjTL5f&noteId=Kv4lOJURpk) and generalization of Elo to multitask we would ask that reviewer Reviewer **NZSD** consider increasing their score to reflect the additional experiments, improvements and clarifications we have provided. We are very happy to continue to engage and answer any questions.

---

> > ### Comment · Reviewer_NZSD · 2024-08-13
> > **Thanks for the response**
> >
> > Thanks for the response.
> >
> > I will keep my original (positive) score.

---

### Official Review · Reviewer_jqvp · 2024-07-30

**Soundness:** 3
**Presentation:** 4
**Contribution:** 3
**Rating:** 7
**Confidence:** 3

**Summary:**

This paper examines  how well the Elo rating system works for evaluating LLMs based on human feedback. The authors first check how stable and reliable Elo ratings are under different settings and point out key factors that affect these ratings. They study how changing the order of matches and adjusting settings like the K-factor can influence the consistency of Elo scores. Using both synthetic and real-world data, the paper shows that Elo ratings become more reliable with more match permutations and the right K-factor adjustments.

The paper also discusses the transitivity of Elo scores, showing that this property doesn't always hold for models with similar performance levels. The authors suggest practical tips for more reliable Elo-based evaluations, like running many permutations for stable scores and adjusting the K-factor based on how different the models' performances are. They highlight the limitations of current Elo-based evaluation methods and propose future research directions, such as considering ties and using more complex data simulations to better mimic real human feedback scenarios.

**Strengths:**

1. Despite Elo rating being a popular method for evaluating LLMs, there is a lack of analysis on its robustness. This paper provides a comprehensive investigation, helping us better understand and implement Elo rating for LLMs.
2. The authors explore the reliability and transitivity of Elo rating under different configurations. Their definitions and formulations are clear and reasonable.
3. Experiments are conducted on both synthetic data and real human feedback, offering valuable insights for future Elo rating applications.

**Weaknesses:**

**Major Issue:**

My biggest concern is the relatively small amount of real data used in the experiments. Only 500 prompts and three model pairs are included, with just one pair having similar performance (Dolly-v2-7b vs. Dolly-v2-12b). Given that Chatbot Arena currently includes dozens of LLMs and has released a large amount of preference data, the volume of real data in this study seems insufficient.

**Minor Issues:**
1. As the authors mentioned, not considering ties appears to simplify the widely used "win/tie/lose" practice.
2. The empirical guidelines provided may have limited applicability since most papers on automatic evaluation of LLMs report win rates rather than Elo ratings. Also, running more than 100 permutations seems costly and lacks practicality.

**Questions:**

1. If I want to automatically evaluate $k$ models to get their final Elo ratings, do you think the size of $k$ will affect the robustness of the Elo rating?
2. Why does Dolly-v2-7b have a slightly higher win rate than Dolly-v2-12b in Table 2?

**Limitations:**

The paper discusses its limitations in Section 8. My personal view on the limitations can be found in the weaknesses mentioned above.

---

> ### Author Rebuttal · Authors · 2024-08-06
>
> We thank Reviewer [ jqvp] for their helpful feedback and observations, noting that our study **“offering valuable insights for future Elo rating applications”**. We thank R  jqvp for noting the comprehensive nature of the evaluation, **“on both synthetic and real human feedback”**, and involving **many ablation variants** such as order of matches and K-Factor. We take this opportunity to address R jqvp concerns and feedback below.
>
> > My biggest concern is the relatively small amount of real data used in the experiments. Only 500 prompts and three model pairs are included, with just one pair having similar performance (Dolly-v2-7b vs. Dolly-v2-12b). Given that Chatbot Arena currently includes dozens of LLMs and has released a large amount of preference data, the volume of real data in this study seems insufficient.
>
> We thank the reviewer for the suggestion and agree that additional data will strengthen our findings. We ran additional experiments during this rebuttal to include this new dataset (see shared response above for results).
>
> > As the authors mentioned, not considering ties appears to simplify the widely used "win/tie/lose" practice.
>
> Thanks for the comment. In the case of a tie, the adjustment in Elo rating points is typically smaller, as ties are considered less informative about the relative strengths of the models. Ties are useful to upper and lower bound the pool of models (with very hard prompts that lead to "both bad" ties and vice versa) and otherwise they slow down the eventual convergence to true rankings.
>
> > The empirical guidelines provided may have limited applicability since most papers on automatic evaluation of LLMs report win rates rather than Elo ratings. Also, running more than 100 permutations seems costly and lacks practicality.
>
> Reporting win rates for model evaluation is only feasible when we have a small number of models to compare because we need to play every two combinations of models against each other. Additionally, to get a reliable signal of performance, they are often run on various benchmarks and a large number of prompts. The advantage of the Elo score is that we can obtain a ranking of a large number of models and position a newcomer model well in this relative ranking with only a few pairs of games instead of covering every head-to-head evaluation. We note that widely used leaderboards such as Chatbot Arena [1] use Elo due to these advantages, so our study is timely at illustrating the limitations.
>
> > If I want to automatically evaluate k models to get their final Elo ratings, do you think the size of k will affect the robustness of the Elo rating?
>
> Yes, we believe so. With a larger k we increase the reliability of the Elo rating because this system has more information to work with. More head-to-head comparisons help smooth out anomalies and provide a more accurate representation of each model's true performance. Elo ratings might converge more slowly because each model must be compared against a larger number of other models. However, once they do converge, the ratings are likely to be more accurate. A large and diverse prompt pool will also stabilize the Elo scores, especially regarding real-world performance. If the prompt pool used for Elo diverges from IRL prompts, the Elo scores will not accurately reflect actual performance.
> We will update section 4 in the manuscript to include this detail.
>
> > Why does Dolly-v2-7b have a slightly higher win rate than Dolly-v2-12b in Table 2?
>
> While surprising, this is consistent with the evaluation performance self-reported by the model authors over 7 benchmarks: OpenBookQA, ARC (easy), WinoGrande, HellaSwag, ARC (challenge), PIQA, and BoolQ. In the model cards, it’s shown that the average performance of the 7B model and 12B model is 0.573 and 0.567 respectively [2].
>
> [1] https://arena.lmsys.org/
> [2] https://huggingface.co/databricks/dolly-v2-7b
>
> ---
>
> We hope that we were able to satisfy the reviewer’s asks by running additional experiments on a much larger dataset and clarifying the details in question. If so, we ask that you consider raising your score to reflect and please do let us know if anything requires further clarification.

---

> > ### Comment · Reviewer_jqvp · 2024-08-09
> > **Thank you for the Response**
> >
> > Thank you for the response and additional experiments. This addressed some of my concerns, so I’ve raised the score to 7. Good luck!

---

> > > ### Author Response · Authors · 2024-08-09
> > >
> > > Thank you for raising your score in light of the manuscript clarifications and the additional experiments. We are glad that we addressed your concerns and are grateful for your valuable suggestions, which allowed us to clarify the significance of our results and run additional experiments to confirm our insights.
> > >
> > > We would like to thank the reviewer again for their positive update and detailed feedback on the presentation of the current manuscript. We would be happy to clarify any other concerns or questions we can address during the rebuttal period.

---

> ### Author Response · Authors · 2024-08-08
>
> Now that the discussion is underway, we wanted to ask Reviewer **jqvp** if there are any follow-up points we can clarify. If there are no further points of clarification regarding the manuscript and size of the dataset (with the new experiments shared here: https://openreview.net/forum?id=Pc9LLjTL5f&noteId=Kv4lOJURpk), the practical applicability of our guidelines, and the impact of the number of models on robustness, we would ask that reviewer Reviewer **jqvp** consider increasing their score to reflect the additional experiments, improvements and clarifications we have provided. We are very happy to continue to engage and answer any questions.

---

### Author Rebuttal · Authors · 2024-08-06

We appreciate the thoughtful and positive feedback from the reviewers.

We are encouraged that the reviewers found the paper comprehensive and helpful and the experimental setup clear and detailed [jqvp, hqxB] **“This paper provides a comprehensive investigation, helping us better understand and implement Elo rating for LLMs**.

We are also glad that the reviewers found the problem we try to address to be well motivated [NZSD, 9SNU]: **“The reliability of the Elo rating is an interesting and meaningful research topic”** and **“The topic studied in this paper (Elo ratings for LLMs) is important, and the experiments are extensive”**

Reviewers also positively commented on the usefulness and clarity of our formulations [jqvp, NZSD] **“Their definitions and formulations are clear and reasonable”** and **“highlight two Axioms for Elo-based evaluation, which is reasonable and useful”**.

We are happy that the rigor and reliability of verifying our findings through both synthetic human feedback and real-world scenarios are appreciated [jqvp, NZSD, hqxB]: **“By combining results from simulated experiments and real-world scenarios, this paper demonstrates good reliability in its findings.”**

We are heartened by the reviewers' excitement about our findings and guidelines for a more robust evaluation of LLMs and acknowledgment of the impact this paper can have [NZSD, 9SNU]:  **“The paper provides several interesting lessons for using Elo ratings for LLMs that I believe can (positively) influence how we compare models in rankings based on pairwise comparisons”.**

---

Below we take this opportunity to clarify some of the shared concerns between reviewers:

**1. On the details of Elo score, in particular, the explanation of the $K$-factor**

[NZSD] and [9SNU] raised valuable points asking for an explanation of the $K$-factor in the Elo formula to be part of the main paper and not in the appendix. The $K$-factor is a crucial parameter that determines how much a player's rating changes after each game and controls the sensitivity of the rating system to new results.
We agree that a more detailed introduction to the Elo score would significantly enhance understanding of the analyses in the paper, as well as clarify the roles of various parameters. We will make changes to the final manuscript to address this concern.

**2. Strengthening the findings with a more comprehensive real-world experiment**

Reviewers raised valid and valuable concerns regarding the potentially low volume of real data in this study [jqvp], asked about increasing the number of models in the pool of comparison [hqxB and NZSD], the possibility of running into lucky sequences when the dataset is small [9SNU].

To address these concerns, we used the opportunity to run new experiments for this rebuttal using the preference dataset released by Chatbot Arena [1] which contains 33k conversations between 20 models and preference annotations from 13k users. In the manuscript, we studied the Elo score robustness to rank models with two changes: variations in the number of permutations and the $K$-factor.

Kindly see the attached PDF for the results of these experiments. Figure A1 shows the pairwise winrate plot of the 20 models in this dataset.


**2.1. Impact of $K$-factor on Elo ranking:**

Out of the 20 models, 19 change their ranking position depending on different values of $K$, with the largest position difference being 6 (observed for llama-13b and alpaca-13b). On average, the rankings shift by 0.94 positions when different values of $K$ are used. Figure A2 in the attached PDF presents the heatmap highlighting this change in the models' rankings.


**2.2. Impact of number of permutations on Elo ranking:**

For a fixed $K$-factor value ($K=4$, as recommended by Chatbot Arena), we study the impact of the order of matches on computing the Elo score. We perform 10 runs of Elo computation, where the only difference is the shuffled order of matches in each run. We observe that the rankings change from one run to the next. Using one run as a reference and comparing the rankings of each shuffled run to this reference, we find that, on average, 11.7 out of the 20 models change positions in the rankings with each shuffling of orders. Figure A3 in the attached PDF presents the results.


When we shift from calculating the Elo score in a single run (the current standard in the field) to performing 10 random permutations of the sequence and averaging the Elo scores, we observe an average difference of 2.9 in rankings. Performing 100 permutations reduces this difference to 0.9 compared to the 10-shuffling run. The difference becomes minimal (0.1) when increasing from 100 to 1000 permutations. This finding is consistent with our observations in the synthetic scenario and the small dataset analyzed in the manuscript. We observe that the number of permutations required to arrive at a reliable and stable Elo ranking depends on the size of the dataset and the distribution of winrates: we achieved consistent Elo scores with 100 permutations on our smaller dataset (ranking 4 models) and 1000 permutations to achieve stability with the Chatbot Arena dataset (ranking 20 models).
See Figure A4 in the attached PDF, which presents the results.




[1] huggingface.co/datasets/lmsys/chatbot_arena_conversations

---

### Decision · Program_Chairs · 2024-09-25

**Decision:**

Accept (poster)

**Comment:**

This paper evaluates the effectiveness of the Elo rating system for assessing large language models (LLMs) using human feedback. The authors investigate the stability and reliability of Elo ratings under various conditions, identifying key factors that impact these scores, such as match order and the K-factor setting. Through experiments with both synthetic and real-world data, they demonstrate that Elo ratings become more consistent with increased match permutations and appropriate K-factor adjustments. Additionally, the paper reveals that Elo score transitivity often fails for models with similar performance, suggesting that Elo-based evaluations can be unreliable in certain cases. To improve reliability, the authors recommend running numerous permutations and adjusting the K-factor according to model performance differences. The paper also addresses the limitations of Elo-based methods and suggests future research directions, including incorporating ties and using more complex simulations to better reflect real-world human feedback.

While the reviews and the author-reviewer discussions are quite extensive, the most critical concern of the two lower reviews are with respect to the writing. Unfortunately, I cannot identify with the authors' argument that this concern is overweighted, since writing is extremely important in scientific communication and the concern was raised by more than one reviewer. I am hence reluctant about this paper and my recommendation is given with a low confidence, even though I am leaning to the positive since the paper has the potential to become quite impactful.